# Na_0.76_V_6_O_15_/Activated Carbon Hybrid Cathode for High-Performance Lithium-Ion Capacitors

**DOI:** 10.3390/ma14010122

**Published:** 2020-12-30

**Authors:** Renwei Lu, Xiaolong Ren, Chong Wang, Changzhen Zhan, Ding Nan, Ruitao Lv, Wanci Shen, Feiyu Kang, Zheng-Hong Huang

**Affiliations:** 1State Key Laboratory of New Ceramics and Fine Processing, School of Materials Science and Engineering, Tsinghua University, Beijing 100084, China; thulrw@163.com (R.L.); 18801291089@163.com (X.R.); wang-c18@mails.tsinghua.edu.cn (C.W.); zcz@gmomi.com (C.Z.); lvruitao@tsinghua.edu.cn (R.L.); shenwc@mail.tsinghua.edu.cn (W.S.); fykang@sz.tsinghua.edu.cn (F.K.); 2College of Chemistry and Chemical Engineering, Inner Mongolia University, Hohhot 010021, China; 3Key Laboratory of Advanced Materials (MOE), School of Materials Science and Engineering, Tsinghua University, Beijing 100084, China

**Keywords:** Na_0.76_V_6_O_15_ nanobelts, activated carbon, hybrid cathode, high electrochemical performance, lithium-ion capacitors

## Abstract

Lithium-ion hybrid capacitors (LICs) are regarded as one of the most promising next generation energy storage devices. Commercial activated carbon materials with low cost and excellent cycling stability are widely used as cathode materials for LICs, however, their low energy density remains a significant challenge for the practical applications of LICs. Herein, Na_0.76_V_6_O_15_ nanobelts (NaVO) were prepared and combined with commercial activated carbon YP50D to form hybrid cathode materials. Credit to the synergism of its capacitive effect and diffusion-controlled faradaic effect, NaVO/C hybrid cathode displays both superior cyclability and enhanced capacity. LICs were assembled with the as-prepared NaVO/C hybrid cathode and artificial graphite anode which was pre-lithiated. Furthermore, 10-NaVO/C//AG LIC delivers a high energy density of 118.9 Wh kg^−1^ at a power density of 220.6 W kg^−1^ and retains 43.7 Wh kg^−1^ even at a high power density of 21,793.0 W kg^−1^. The LIC can also maintain long-term cycling stability with capacitance retention of approximately 70% after 5000 cycles at 1 A g^−1^. Accordingly, hybrid cathodes composed of commercial activated carbon and a small amount of high energy battery-type materials are expected to be a candidate for low-cost advanced LICs with both high energy density and power density.

## 1. Introduction

Due to the rapid development of electronic devices, energy storage systems with both high energy and power density have attracted tremendous attention in recent years [1,2,3]. Due to the mechanism of Li ions intercalation/deintercalation into electrode materials, lithium-ion batteries (LIBs) deliver a high energy density and are widely used in our daily life. However, its low power density and poor cycling stability limit its further application [4,5,6]. For supercapacitors (SCs), ultrahigh power density and superior cycle life are achieved by the mechanism of reversible physical adsorption/desorption of ions at the surface of the electrode materials, but relatively low energy density is unavoidable [7,8,9]. Lithium-ion capacitors (LICs) with a battery-type anode and a capacitor-type cathode were invented to build the bridge between the LIBs and SCs [10,11]. Benefitting from the outstanding rate capability of cathodes and the high capacity of anodes, LICs can work at both high-power and high-energy modes. To date, plenty of advanced LICs with balanced energy and power performances have been published, indicating a promising energy storage device aimed towards widespread practical application [12,13,14,15,16].

Benefitting from its low electrode potential of around 0 V compared to Li/Li^+^, carbon-based materials, including artificial graphite, natural flake graphite and hard carbon, have been widely developed as the anodes for LICs [17,18,19]. The insertion/extraction process of Li ions provides high capacity and outstanding rate performance. Currently, advanced porous carbon materials with optimized pore-structures and high specific surface areas show excellent performance as the cathodes for LICs. Hao et al. synthesized sponge-like carbon with a large specific surface area of 2651 m^2^ g^−1^ and used it as both the cathode and the anode of the LIC, achieving the maximum energy density of 127 Wh kg^−1^ and the maximum power density of 33,573 W kg^−1^ with a capacitance retention over 99% after 100,000 cycles [20]. Wang et al. assembled an LIC device with an activated 3D graphene sheet cathode, delivering a high energy density of 111.4 Wh kg^−1^ at 11,250 W kg^−1^ and showing 6.5% capacitance fade after 3000 cycles at a current density of 2 A g^−1^ [21]. Palanichamy et al. reported an all carbon-based LIC composed of a hard carbon anode and an activated carbon cathode, showing an ultra-high energy density of 216 Wh kg^−1^ and a remarkable cycling stability of ~94% initial capacity after 5000 cycles [22]. Nevertheless, it remains challenging to achieve the large-scale application of these advanced porous carbon materials, owing to some insurmountable drawbacks such as the high costs, low yields and complex technique.

Low-cost commercial activated carbon materials with developed micropore structure show great cyclability and excellent power performance when employed as the cathode materials for LICs, indicating an ideal candidate for LICs if its relatively low energy density is ignored. Therefore, facile and effective approaches to enhance the energy density of commercial activated carbon are of great significance to promote the widespread application of high-performance LICs. In this work, a hybrid cathode was designed by combining commercial activated carbon YP50D with battery-type component Na_0.76_V_6_O_15_ nanobelts (NaVO). NaVO materials were prepared by a facile hydrothermal method and subsequent annealing according to Yang’s report [23]. Due to the combination of two kinds of Li ions storage mechanisms, NaVO/C hybrids show both high specific capacity and superior cycling stability. LIC was fabricated with a 10-NaVO/C hybrid cathode and an artificial graphite anode after pre-lithiation, delivering a high energy density of 118.9 Wh kg^−1^ at a power density of 220.6 W kg^−1^ and maintaining 43.7 Wh kg^−1^ at a high power density of 21,793.0 W kg^−1^. A long cycle life of 5000 cycles with capacitance retention of approximately to 70% was obtained at 1 A g^−1^. These results demonstrated the great possibility of combining commercial activated carbon and small amounts of high energy battery materials uniformly to prepare high-performance LIC hybrid cathodes with low cost and high yields.

## 2. Experimental

### 2.1. Material Synthesis

Figure 1 illustrates the preparation procedure of the Na_0.76_V_6_O_15_/YP50D hybrids. NaCl and sodium dodecyl sulfate (SDS) were added into NH_4_VO_3_ solution, and then, a typical hydrothermal method at 200 °C was carried out to obtain the precursors. After annealing at 400 °C for 4 h, Na_0.76_V_6_O_15_ nanobelts (NaVO) were successfully synthesized [23]. Then, NaVO was mixed uniformly with commercial activated carbon YP50D (Kuraray, Tokyo, Japan) under vigorous stirring and the samples were marked as 10-NaVO/C, 20-NaVO/C and 30-NaVO/C according to the mass percentages of NaVO, respectively.

### 2.2. Characterization

A GeminiSEM500 (Carl Zeiss, Oberkochen, Germany) was used at 15 kV to collect scanning electron microscope (SEM) images. A JEM-2100F (JEOL, Tokyo, Japan) was operated at 200 kV to collect transmission electron microscope (TEM) images and further energy dispersive spectroscopy (EDS). An X-ray diffractometer (XRD) D/max-2500/PC (Rigaku, Tokyo, Japan) with Cu Kα radiation was used to record X-ray diffraction data. Nitrogen adsorption method was used to investigate the pore volume, pore size distribution and specific surface area, using Quadrasorb SI-MP adsorption apparatus (Quantachrome, Boynton Beach, FL, USA) at 77 K. A LabRAM HR Evolution system (Horiba, Kyoto, Japan) was employed to carry out Raman spectroscopy with 532 nm excitation wavelength.

### 2.3. Electrochemical Measurements

All the electrochemical measurements were carried out at room temperature. Active materials (YP50D, 10-NaVO/C, 20-NaVO/C or 30-NaVO/C), polyvinylidene fluoride (PVDF) and acetylene carbon black (Super P) with a mass ratio of 8:1:1 were mixed and dispersed in *N-*methyl-2-pyrrolidone (NMP) to form the uniform slurry. Then, the slurry was coated onto a carbon-coated Al foil and dried at 120 °C in a vacuum drying oven for 8 h, and used as the cathode electrodes for half-cell and LIC devices testing. Pure Li metal foil was used as the counter electrodes in the half-cell testing. The anode electrodes for lithium-ion hybrid capacitors were prepared by 70% artificial graphite powder, 15% PVDF and 15% Super P using the same procedure as the cathode electrodes, followed by a pre-lithiation process. All the electrodes were 12 mm diameter pieces, with an active mass loading of 1–5 mg cm^−2^. Then CR2032 coin cells were fabricated inside a glovebox in Ar atmosphere (the contents of both H_2_O and O_2_ are less than 0.1 ppm), using 1 M LiPF_6_ in ethylene carbonate/dimethyl carbonate (EC/DMC) (1:1, by volume) as electrolyte and polypropylene membrane as separators.

SCTS Arbin instruments were used to carry out the cyclic voltammetry (CV) tests of the LIB half-cells (1.5–4 V) and the LIC devices (1–3.8 V). The galvanostatic charge/discharge experiments on the LIB half-cells between 1.5 and 4 V were carried out using a LAND multichannel battery tester CT2001A. Electrochemical measurements for the LIC devices were performed using an Arbin-BT2000 test station in a voltage window of 1–3.8 V. Discharge energy (*e_D_*, Wh) and discharge time (*t_D_*, s) were measured to calculate energy density (E, Wh kg^−1^) and power density (P, W kg^−1^) based on the following equations:(1)$$E=\frac{e_{D}}{m}\times 1000$$
(2)$$P=\frac{E\times 3600}{t_{D}}$$
where *m* (kg) is calculated by adding the mass of active substances in both electrodes together.

## 3. Results and Discussion

### 3.1. Morphology and Physicochemical Characterization

The XRD patterns of the samples are displayed in Figure 2a. All the samples exhibit the same sharp peaks as pure NaVO, indexed to the monoclinic crystalline phase of Na_0.76_V_6_O_15_ (space group: C2/m, JCPDS Card No. 75-1653). In addition, the XRD patterns of NaVO/C hybrids show two broad diffraction peaks at approximately 24° and 44°, corresponding to the (002) and (100) diffraction of the amorphous structure of YP50D. No other peaks were observed, demonstrating the high purity of the samples. Figure 2b–d and Appendix A show the SEM images of the samples. NaVO is a cluster of one-dimensional nanobelts with average width of ~180 nm, possessing smooth surface and relatively uniform orientation (Figure 2b). As shown in Figure 2c, YP50D are particles with an average size of several microns. It is distinct that NaVO nanobelts and activated carbon YP50D particles are well mixed (Figure 2d). High Resolution Transmission Electron Microscope (HR-TEM) and selected area electron diffraction (SAED) are performed to further investigate the details of the structure (Figure 2e). The interlayer spacing of d = 3.40 Å corresponds to the (−111) crystal plane of Na_0.76_V_6_O_15_, which is consistent with the XRD results. The SAED confirms the high crystallinity of NaVO nanobelts. Appendix A shows uniform distribution of Na, V, O for the energy dispersive X-ray spectroscopy mapping.

Raman measurements were carried out to further study the phase composition of the samples (Figure 2f). For YP50D, a D-band located at 1331 cm^−1^ and a G-band at 1590 cm^−1^ correspond to the disordered graphitic structure and the graphitic structure, respectively [24,25]. The high intensity ratio (*I_D_*/*I_G_* ≈ 2.03) calculated by integrate indicates the amorphous phase of YP50D. For NaVO, the phono mode at 337 cm^−1^ is assigned to the bond bending vibrations, and the 448 cm^−1^ band corresponds to both the V-O stretching vibrations and the V-O-V bending vibrations. The other modes at 518, 673 and 1002 cm^−1^ originate from the stretching vibrations of different V-O bonds [26,27,28]. N_2_ adsorption-desorption measurements at 77 K were undertaken to investigate the specific surface area and porosity of YP50D. The specific surface area of YP50D is 1411 m^2^ g^−1^. As shown in Appendix A, the adsorption-desorption curve exhibits a standard type-I isotherm and the adsorption volume increases rapidly below P/P_0_ = 0.01, indicating the dominant micropore structure. The pore size distribution profile was shown in Appendix A. The sharp peak at approximately 1 nm was observed, indicating that the micropore structure is preponderant. Overall, all the morphology and physicochemical characterization confirms that Na_0.76_V_6_O_15_ nanobelts were successfully synthesized and mixed uniformly with activated carbon YP50D in different proportions.

### 3.2. Electrochemical Performance as the Cathode in the Half-Cell

Lithium ion half-cells with lithium metal counter electrodes were fabricated to measure the electrochemical performances of the samples as the cathode materials. Figure 3a shows the CV curves of YP50D and the NaVO/C hybrids at a scan rate of 0.5 mV s^−1^. The CV curve of YP50D has a standard rectangular shape, while all the CV curves of the NaVO/C hybrids show three apparent reduction peaks located at approximately 3.25 V, 2.86 V and 2.37 V (vs. Li/Li^+^), ascribed to the multi-step intercalation of lithium ions [23,29]. A solid solution transformation contributes to the peak at 3.25 V, the reduction of vanadium from V^5+^ to V^4+^ and V^4+^ to V^3+^ leads to the peaks at 2.86 V and 2.37 V, respectively [23]. Two anodic peaks located at 3.03 V and 3.05 V (vs. Li/Li^+^) are observed as well, corresponding to the de-intercalation of lithium ions. The whole intercalation/de-intercalation process of lithium ions can be summarized by the following equation:(3)$${\mathrm{Na}}_{0.76}V_{6}O_{15}+{x\;\mathrm{Li}}^{+}+{\mathrm{xe}}^{-}\overset{\;}{↔}{\mathrm{Li}}_{x}{\mathrm{Na}}_{0.76}V_{6}O_{15}\;$$

In addition, the CV curves show larger areas when the content of NaVO increases, indicating that the increased redox reactions deliver a higher capacity. Figure 3b and Appendix A shows the CV curves of all the samples at various scan rates (from 1 to 50 mV s^−1^). With the increase of the scan rate, the redox peaks shift slightly and the shape of CV curves is well retained, demonstrating excellent reaction kinetics and outstanding rate performances. Moreover, the CV curves of 10-NaVO/C at large current densities are quasi-rectangular with unobvious redox peaks, which are similar to the CV curves of YP50D, indicating that capacitive behavior dominates and battery behavior contributes to capacity as well.

To further investigate the charge/discharge mechanism of the hybrid cathodes, the contribution ratios of both the diffusion-controlled faradaic process and the capacitive process are analyzed based on the following formula:(4)$$i\left(V\right)=k_{1}v+k_{2}v^{0.5}\;$$
where i(V) is the testing current at a certain potential, v is the measured scan rate (mV s^−1^), k_1_v and k_2_v^0.5^ are indexes corresponding to the ratios of the capacitive process and the diffusion-controlled faradaic process, respectively [30,31,32]. The capacitive contribution profiles at the scan rate of 1 mV s^−1^ are shown in Appendix A. It is obvious that the diffusion-controlled contribution of 10-NaVO/C mainly centers at the redox peaks. Figure 3c compares the contribution ratios of two kinds of processes for YP50D and 10-NaVO/C. The faradaic process contribution ratios of both materials decreases when the scan rate increases from 1 to 20 mV s^−1^, but the ratio of 10-NaVO/C hybrid is higher than that of YP50D all the time, demonstrating that the battery-type material NaVO contributes extra capacity throughout. It is remarkable that for 10-NaVO/C capacitive behavior plays a dominant role at all scan rates, contributing to excellent cycling stability and superior rate capability, and the more diffusion-controlled faradaic process enhances specific capacity. The GCD profiles of all the samples at 0.05 A g^−1^ are shown in Figure 3d. The non-linear curves of 20-NaVO/C and 30-NaVO/C with apparent charge-discharge plateaus represent larger ratios of battery-type behavior; while 10-NaVO/C displays a quasi-line profile, related to a synergistic effect of dominant capacitive process and some faradaic process, which is consistent with the previous kinetic analysis.

The excellent rate performance of NaVO/C is shown in Figure 3e. For three kinds of NaVO/C samples, high specific charge/discharge capacities of 84.8, 97.5 and 116.5 mAh g^−1^ at a small current density of 0.05 A g^−1^ and 51.4, 51.4 and 54.2 mAh g^−1^ at a large current density of 5 A g^−1^ are achieved, respectively. The higher capacity of NaVO/C than YP50D is mainly due to the combined behavior of the capacitive and the faradaic processes. The initial coulombic efficiencies are 99.2%, 97.5% and 97.7% for 10-NaVO/C, 20-NaVO/C and 30-NaVO/C, respectively. Meanwhile, as Figure 3f shows, 10-NaVO/C displays an outstanding cycling stability. The NaVO/C samples deliver an initial capacity of 70.3 mAh g^−1^, 73.6 mAh g^−1^ and 61.1 mA h g^−1^ at a current density of 1 A g^−1^, respectively, higher than the 53.2 mAh g^−1^ of YP50D. After 2000 cycles, 10-NaVO/C still maintains a high specific capacity of 56.4 mAh g^−1^ with a significant small fade, indicating an ideal cathode material. The comprehensive electrochemical performances of the 10-NaVO/C hybrid cathode are mainly attributed to the synergism of two types of cathode materials. Battery-type NaVO promotes specific capacity, and capacitor-type YP50D ensures outstanding rate capability and long cycle life span. In summary, the 10-NaVO/C hybrid cathode with combined capacitive and faradaic behavior is expected to be a candidate for high-performance lithium-ion hybrid capacitors.

### 3.3. Electrochemical Performances of Full Lithium-Ion Capacitor Devices

The 10-NaVO/C hybrid cathode and the commercial artificial graphite (AG) anode after pre-lithiation were prepared to fabricate the LICs, using 1 M LiPF_6_ in EC/DMC (1:1, by volume) as electrolyte. The rate and cycling performance of AG is displayed in Appendix A. The pre-lithiation process for the graphite anode was carried out by galvanostatic charging/discharging at a small current density of 0.1 A g^−1^ for two cycles and ending at about 0.1 V to reduce the irreversible reactions and form stable solid electrolyte interface (SEI) film. Considering the working potential of the cathode and the anode, the potential window of 1–3.8 V was chosen for the 10-NaVO/C//AG LIC devices to obtain the best electrochemical performances.

Figure 4a and Appendix A show the CV curves of the 10-NaVO/C//AG and YP50D//AG LICs at the potential window of 1–3.8 V. The CV curves of 10-NaVO/C//AG display a quasi-rectangular shape with several redox peaks, indicating a hybrid storage mechanism occurring in the device. During the charging procedure, PF_6_^–^ ions are adsorbed into the advanced pore structure of the hybrid cathode (capacitive process) and Li^+^ ions embedded into the layer structure of the anode (faradaic process). In addition, the oxidation reactions of vanadium ions in NaVO occurred (faradaic process), enhancing the energy density [23,31]. Completely reversible reactions take place in the discharging process. The well-retained shape when the scan rate increases from 1 to 50 mV s^−1^ indicates the superior reversibility and outstanding rate performance. The GCD profiles of 10-NaVO/C//AG and YP50D//AG devices are shown in Figure 4b and Appendix A. The non-linear shape of the curves measured for the 10-NaVO/C//AG LIC device further confirms the “coupling effect” of the capacitive process and faradaic process in the device, consistent with the above CV curves. Meanwhile, the symmetrical profiles demonstrate the ultra-high coulombic efficiency and extraordinary reversibility of the LICs.

It is well known that the capacity balance between the cathode and anode plays a significant role in optimizing the electrochemical properties of LIC devices [33], so four mass loading ratios of the cathode to anode from 1:1 to 2.5:1 were investigated. As shown in Figure 4c, the LICs with 1.5:1 and 2:1 cathode-to-anode mass ratios show higher rate capability, with a specific capacitance of 67.3, 55.8 F g^−1^ at 0.1 A g^−1^ and 25.1, 28.3 F g^−1^ at a large current density of 10 A g^−1^, respectively. Figure 4d and Appendix A display the cycling performance of the two devices, the 10-NaVO/C//AG (1.5:1) device achieves an outstanding cyclability of capacitance retention of approximately 70% after 5000 cycles at 1 A g^−1^, with a superior coulombic efficiency of nearly 100%. Appendix A shows the Ragone plots of 10-NaVO/C//AG with various cathode-to-anode weight ratios, in which energy density and power density were calculated according to Equations (1) and (2). Among all the LIC devices, the 10-NaVO/C//AG (1.5:1) device shows the most outstanding electrochemical performance, delivering a high energy density of 118.9 Wh kg^−1^ at a power density of 220.6 W kg^−1^ and 43.7 Wh kg^−1^ even at 21,793.0 W kg^−1^. The 10-NaVO/C//AG (1.5:1) device performs better than the YP50D//AG LIC device and several previously reported works as shown in in Figure 4e [34,35,36,37,38,39]. Furthermore, a yellow LED was successfully lit by a full-charged 10-NaVO/C//AG (1.5:1) device, indicating the great potential in practical applications (Figure 4f).

## 4. Conclusions

In this work, Na_0.76_V_6_O_15_ nanobelts were synthesized via a typical hydrothermal method and subsequent annealing to enhance the energy density of commercial activated carbon when employed as the cathode for lithium-ion hybrid capacitors. Benefiting from the synergism of the capacitive process and the diffusion-controlled faradaic process, the hybrid cathode 10-NaVO/C shows both high capacity and outstanding cycling stability. LICs were assembled using the 10-NaVO/C hybrid cathode and the artificial graphite anode after pre-lithiation, maintaining a high energy density of 118.9 Wh kg^−1^ at a power density of 220.6 W kg^−1^ and 43.7 Wh kg^−1^ even at an ultra-high power density of 21,793.0 W kg^−1^. Superior cycling stability of the capacitance retention of approximately 70% was achieved after 5000 cycles at a current density of 1 A g^−1^. Above results confirm 10-NaVO/C as a potential candidate cathode material for high-performance LICs. Meanwhile, the demonstrated energy and power density prove the possibility of using hybrid cathodes composed of commercial activated carbon and small amounts of high energy battery-type materials to prepare advanced energy storage devices with low cost.

## Figures and Tables

**Figure 1 materials-14-00122-f001:**
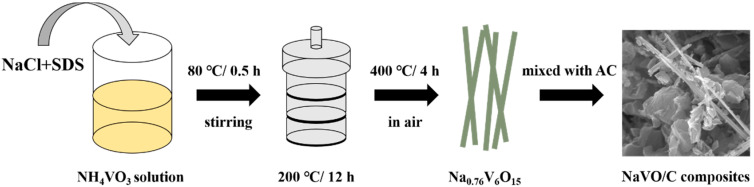
Synthesis process of the NaVO/C hybrids.

**Figure 2 materials-14-00122-f002:**
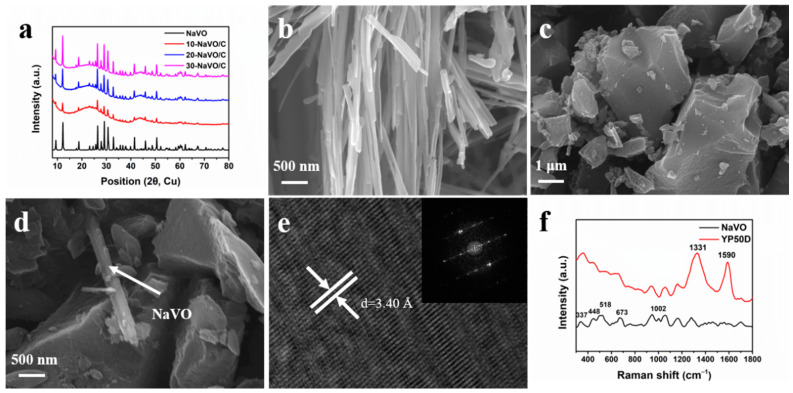
(**a**) XRD patterns of NaVO and the NaVO/C hybrids; (**b**–**d**) SEM images of NaVO nanobelts, YP50D and 10-NaVO/C; (**e**) HR-TEM micrograph of NaVO nanobelts (inset shows corresponding SAED); (**f**) Raman spectra of NaVO and YP50D.

**Figure 3 materials-14-00122-f003:**
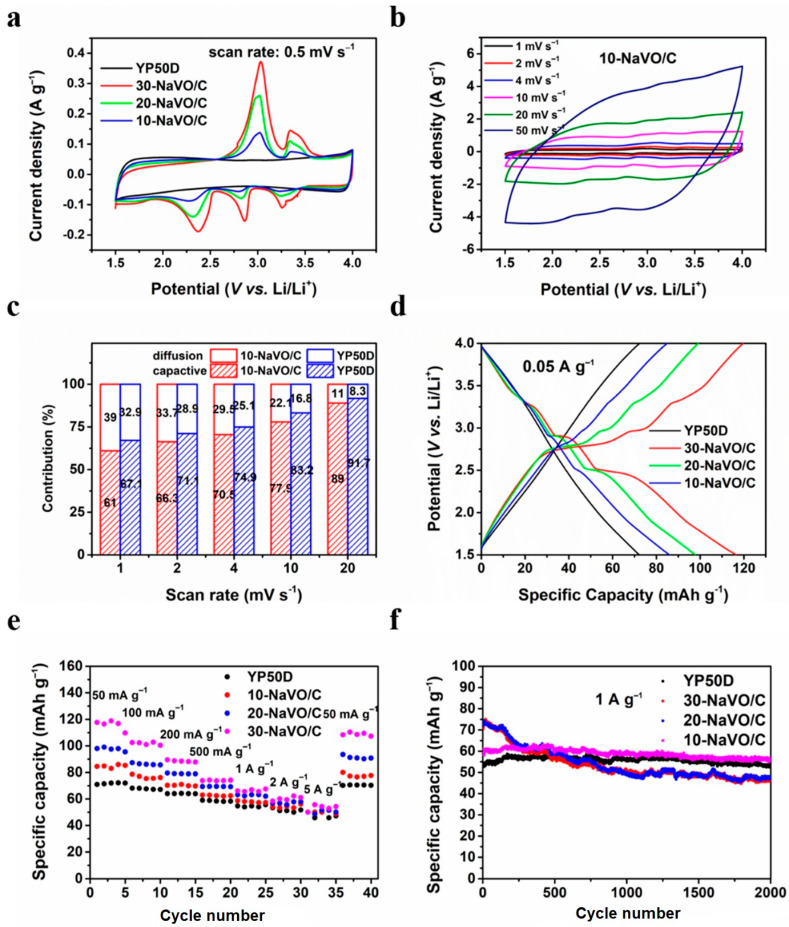
Electrochemical performances of YP50D and the NaVO/C hybrids in half-cell tests from 1.5 to 4 V. (**a**) Cyclic voltammetry (CV) curves of all the samples at a scan rate of 0.5 mV s^−1^; (**b**) CV curves of 10-NaVO/C at different scan rates between 1 and 50 mV s^−1^; (**c**) contribution ratios of the diffusion-controlled faradaic process and capacitive process for 10-NaVO/C; (**d**) galvanostatic charge/discharge (GCD) profiles of all the samples at a current density of 0.05 A g^−1^; (**e**) rate performance at various current densities; (**f**) cycling behavior at 1 A g^−1^ for 2000 cycles.

**Figure 4 materials-14-00122-f004:**
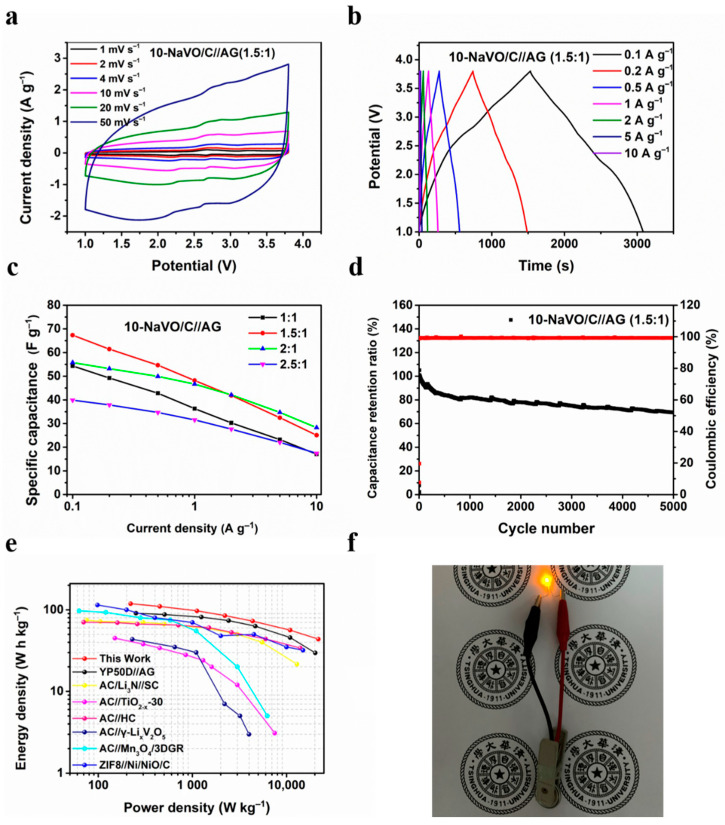
Electrochemical performances of 10-NaVO/C in Li-ion capacitor tests from 1 to 3.8 V. (**a**) CV curves of the 10-NaVO/C//AG (1.5:1) lithium-ion hybrid capacitor (LIC) device at various scan rates; (**b**) charge/discharge potential profiles at different current densities; (**c**) specific capacitance of the LICs with various cathode-to-anode weight ratios; (**d**) cycle behavior of the 10-NaVO/C//AG (1.5:1) device at a current density of 1 A g^−1^; (**e**) Ragone plots of the 10-NaVO/C//AG (1.5:1) device and other advanced LICs in previous work; (**f**) digital photograph of the lightened yellow LED.

## Data Availability

The article or Appendix A include all data.

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
