# Peer review of "Na0.76V6O15/Activated Carbon Hybrid Cathode for High-Performance Lithium-Ion Capacitors"

_materials, 2020, doi:10.3390/ma14010122_

Round 1

Reviewer 1 Report

The authors suggest a hybrid cathode composed of commercially-available activated carbon with battery-type components. Thanks to a dual way Li-ion can be stored, the suggested hybrid cathode achieves both good specific capacity and cycling stability. This work serves as a proof of concept for hybrid cathode materials composed of activated carbon and high-energy battery materials.

The work is interesting, decently presented and pertinent, in scope, to MAterials. Thus I recommend publication of the manuscript once the few comments I have below are addressed.

- Some figures (for instance Figure 3) are nearly impossible to read. Please consider enlarging them. A good rule of thumb is that the written part of the graphic should have the same font and font size as the rest of the text.

- What is the temperature dependence of their observation? All the electrochemical characterisation is done at room temperature.

- How the authors speculate their results will be used for further development of the technology?

Reviewer 2 Report

Na0.76V6O15 nanobelts (NaVO) were prepared and combined with commercial activated carbon YP50D to form hybrid cathode materials.

It showed good electrochemical performance.

However, I recommend to reply the following questions before the publication.

1. It was suggested that the reduction of sodium from Na+ to Na0 contributes to the peak at 3.25 V.

However, redox potential of Sodium is close to Li. Please explain that in detail.

2. Is there any concern to move sodium ion instead of lithium ion?

3. It is better to check the redox reaction of vanadium by XPS or XAFS. 
